# Current Approaches to Use Cyclodextrins and Mucoadhesive Polymers in Ocular Drug Delivery—A Mini-Review

**Tivadar Bíró and Zoltán Aigner ***

University of Szeged, Institute of Pharmaceutical Technology and Regulatory Affairs, Eötvös u. 6.
H-6720 Szeged, Hungary
* Correspondence: aigner@pharm.u-szeged.hu; Tel.: +36-62-545-577

**Abstract:** Ocular drug delivery provides a challenging opportunity to develop optimal formulations with proper therapeutic effects and acceptable patient compliance because there are many restricting factors involved, such as complex anatomical structures, defensive mechanisms, rapid drainage, and applicability issues. Fortunately, recent advances in the field mean that these problems can be overcome through the formulation of innovative ophthalmic products. Through the addition of solubility enhancer cyclodextrin derivatives and mucoadhesive polymers, the permeability of active ingredients is improved, and retention time is increased in the ocular surface. Therefore, preferable efficacy and bioavailability can be achieved. In this short review, the authors describe the theoretical background, technological possibilities, and the current approaches in the field of ophthalmology.

**Keywords:** ocular drug delivery; pharmaceutical technology; ophthalmic formulation; cyclodextrin; mucoadhesion; polymers

## 1. Introduction

Eyes are one of the most important organs of the human body and in the case of any dysfunction in vision, serious drawbacks can appear in daily activities. Ocular drug delivery is a major challenge in the pharmaceutical research and development field because of restrictions caused by many factors. When considering patient-oriented therapy, patient compliance is a key factor, thus the mission of many researchers is to find an optimal administration method that is self-applicable for the patients, and to find the optimal formulation with accomplished therapeutic effect and zero irritation.

### 1.1. Anatomical and Physiological Perspectives

The complex anatomy of the eye limits the amount of therapy options for different diseases, especially when deeper drug permeation is needed. The eye has two main parts; the anterior segment, which includes the cornea, aqueous humor, iris, and lens, and the posterior segment, which includes from the lens to the deeper tissues (vitreous humor, retina, sclera, optic nerve) (Figure 1). The cornea consists of five layers; the lipophilic epithelium with tight junctions, Descemet's membrane, the hydrophilic stroma (which is the thickest part of the cornea), Bowman's layer, and the lipophilic endothelium [1–4]. When considering the optimal drug penetration through the cornea, a balance between the hydrophilicity and lipophilicity of the drug and the delivery system is necessary. Due to its complex anatomical structure, formed physiological barriers protect the eye from surrounding exposures. The first barrier is built by the tear film and includes a lipid layer, mucins, and water. It protects the cornea and conjunctiva. The composition of the corneal barrier was mentioned before. It mainly restricts drug permeation to the anterior tissues. The conjunctival barrier consists of epithelial

layers and connective tissue with blood and lymphatic vessels. The blood–aqueous barrier (BAB) contains tight junctions of the capillary endothelium of the iris, and ciliary epithelium. It is mildly permeable for low-weight molecules. The drug permeation is restricted from systemic circulation to the posterior segment of the eye by the blood–retinal barrier (BRB) due to the tight junctions of retinal pigment epithelium and the endothelial membrane of retinal blood vessels [5–7]. After any stimulus reflex mechanisms, like lachrymal secretion and eye blinking, are induced, thus eliminating the irritative agents in minutes from the eye surface. If the drug is passed through the cornea, the opposite flow of aqueous humor also limits penetration to the posterior direction [8]. Therefore, these mechanisms also limit the therapy by blocking drug permeation into the targeted tissues.

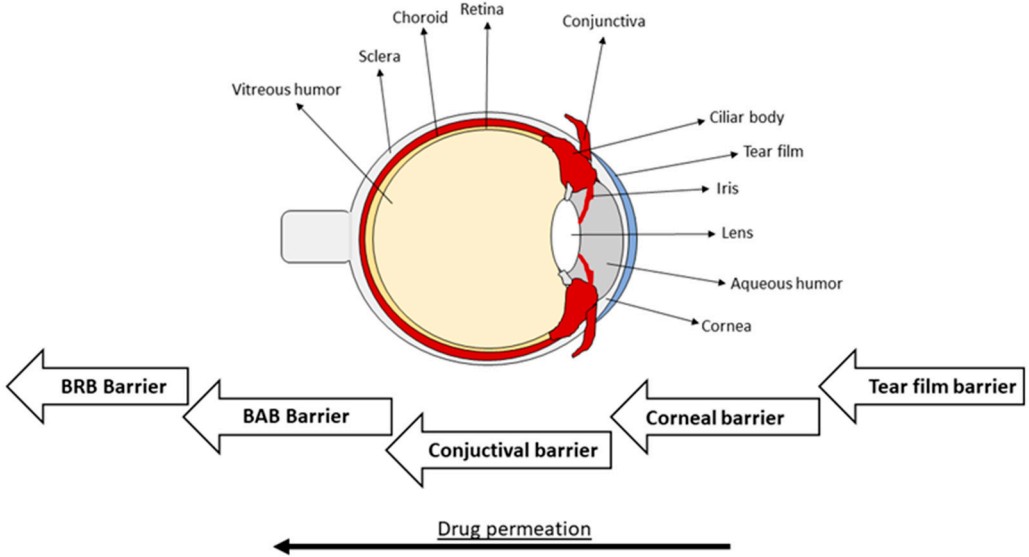

**Figure 1.** Structure of eye and physiological barriers (BRB: Blood–retinal barrier, BAB: Blood–aqueous barrier).

## 1.2. Conventional Routes of Administration

Considering the above-mentioned defensive blockade, the process of ensuring the required therapy is excessively difficult in ophthalmology. In clinical practice, there are invasive and non-invasive methods for administration of the formulation to reach the targeted site (Table 1). Non-invasive routes, also known as topical formulations, are mainly for reaching the anterior segment. The indications at this site are cataract, glaucoma, dry eye, inflammatory diseases, trauma or surgery induced diseases, injury, and tumor. Topical administration is the easiest and most commonly applied non-invasive method which is self-applicable for the patient. Mostly eye-drops, semisolid formulations, inserts, and contact lenses containing the active pharmaceutical ingredient (API) are used [1,9,10]. The requirements are exact for these products. Sterile, isotonic, and microbiologically stable formulations must be prepared with acceptable pH and viscosity. If any of the parameters differ from the optimal range, defensive mechanisms are induced in the eye, therefore the expected efficacy would be much worse. After application, the tear film barrier is the first blockade. For optimal drug permeation, sufficient concentration of drug must present at the cornea. The second obstacle is the corneal barrier, where firstly the drug meets the corneal epithelial multilayer. Because of the tight junction proteins between the epithelial cells, penetration of hydrophilic molecules are restricted, and lipophilic drugs can permeate transcellular by passive diffusion. The second part of the cornea is the stroma, which is a hydrophilic environment, therefore the penetration of lipophilic drugs is restricted there. The lipophilic endothelial monolayer is more transparent for macromolecules than the epithelium. The non-corneal pathway is also known as conjunctival–scleral route, where the permeation mostly depends on the molecular weight. Through the corneal and non-corneal pathway, the anterior tissues are partly

reachable for the active ingredients. From the precorneal area (tear film), the applied formulation is eliminated through the tear turnover and nasolacrimal drainage to the systemic circulation [11–13].

The invasive administration methods like intravitreal, subconjunctival injections, and inserts have both advantages and limitations. With these methods, the target tissues are directly reachable, although these invasive administrations are limited because of the required expertise, proper dosage, and possible side-effects, like toxic reactions of the cornea. The subconjunctival application is less invasive, although the elimination is decently fast through the conjunctival blood and lymphatic vessels.

Oral and intravenous administration are rather unfavored because of the presence of the BAB and the first pass metabolism. To overcome the barrier, a high concentration of the drug needs to be used, which is difficult because of the possible side effects and poor solubility of the most active ingredients [14,15].

**Table 1.** Routes of administration to the eye with advantages and limitations.

| Route of Administration | Advantages | Limitations |
|---|---|---|
| Topical | Patient-compliance, self-applicable, non-invasive, simple, no first-pass effect | Frequent administration needed, low bioavailability, short contact time on the eye surface, tear dilution |
| Subconjunctival | Barely invasive, high efficacy, no first pass effect | Fast clearance, expertise needed, not self-applicable |
| Intravitreal | High bioavailability, avoiding cornea, no first-pass effect | Critical dosing, very invasive method, expertise needed, not patient compliant, toxic side effects |
| Intravenous | Avoiding cornea, less frequent application | Invasive, expertise needed, not targeted exposure, large dose needed |
| Oral | Patent compliant, non-invasive | First pass effect, low ocular efficacy, not targeted exposure, large dose needed |

## 2. Novel Approaches in the Research of Ophthalmic Formulations

When considering the attributes of physiological obstacles of administration routes, an innovative solution is required, one that is acceptable in terms of patient compliance and efficient therapy. A topically self-administrable formulation would be optimal, with enhanced drug permeability into the anterior/posterior tissues and increased residence time on the surface of the eye. Enticing results are published on the impact of complex of drug-cyclodextrin derivatives, mucoadhesive polymers, and nanotechnology. This short review on recent ocular drug delivery approaches summarizes these innovations from recent years.

### 2.1. Cyclodextrins

To reach optimal penetration, the API needs to be dissolved in lachrymal fluid and pass the tear film barrier. If the concentration of API is going to be optimal near the corneal epithelium, a steady amount needs to be ensured for optimal permeation [16]. A major challenge is presented by the fact that the applied APIs in ophthalmology are mostly lipophilic molecules, with low water solubility. Application of solubility enhancer additives, like cyclodextrins (CD) could be the first step for the optimization of eye drop formulations. CDs are cyclic oligosaccharides with α-(1,4) linked α-D-glucopyranose units. In nature, three types are formed by bacterial digestion of starch; α-CD with 6, β-CD with 7, and γ-CD with 8 glucopyranose units. The external surface of these molecules is hydrophilic due to the orientation of hydroxyl groups, which form hydrogen bonds with surrounding water molecules. Inside the cavity of CDs, the environment is hydrophobic, therefore an inclusion complex can be formed with lipophilic agents by hydrogen bonds, van der Waals, and charge-transfer interactions. In aqueous solution dynamic equilibrium is created between the free CD and drug

molecules and the complex. After application on the eye surface, only the free lipophilic molecule can permeate through the cornea, the hydrophilic CD remains and is eventually eliminated through the nasolacrimal pathway (Figure 2).

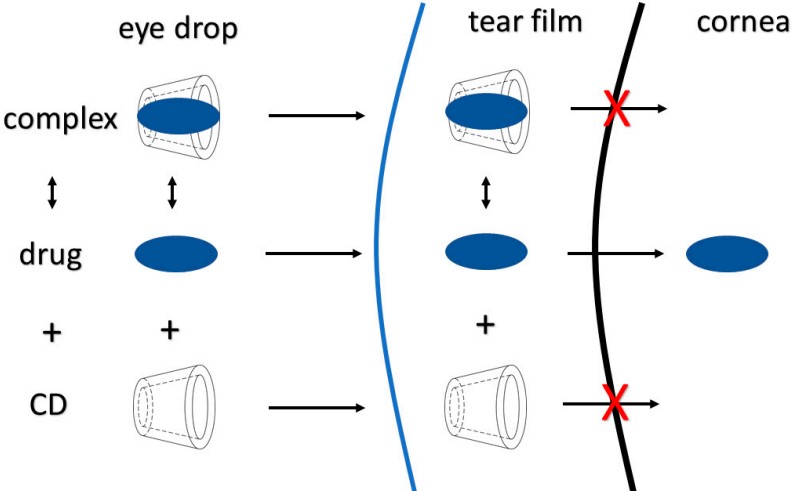

**Figure 2.** Schematic figure about the cyclodextrin drug permeability enhancer attributes.

With the formation of inclusion complexes, the APIs are dissolved in the tear and concentrated near the cornea epithelium. Low or unnecessarily high amounts of CD restrict the permeation of drug, thus the CD concentration needs to be optimized in the formulation [16–21]. To investigate the formation of the inclusion complex in solution, the phase solubility test is a well-known method, which has been described previously by Higuchi and Connors. The stability constant of the complex ($K_S$) is calculable from the slope of phase solubility diagram using Higuchi–Connors equation (Equation (1)):

$$K_s = Slope/\{Intercept\ (1-Slope)\} \qquad (1)$$

The intensity of binding forces between the API and CD molecules can be established by the stability constant [22,23]. A phase–solubility diagram is shown on Figure 3, which was published before by our research group [24].

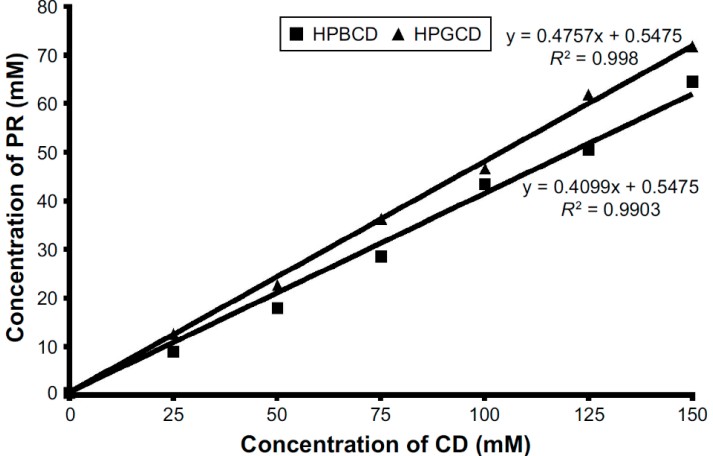

**Figure 3.** Phase–solubility diagram of prednisolone (PR)-hydroxypropyl-β-cyclodextrin (HPBCD) and PR-hydroxypropyl-γ-cyclodextrin (HPGCD) inclusion complexes. The solubility of PR increased linearly by increasing the concentration of cyclodextrins (CDs). The curve is $A_L$ type, thus inclusion complexes with 1:1 molecular ratio are formed in the case of HPGCD and HPBCD [24].

In the solid state, the formation of drug–CD inclusion complexes can be investigated by differential scanning calorimetry (DSC), Fourier-transformed infrared spectroscopy (FTIR), X-Ray powder diffraction (XRPD) and $C_{13}$-NMR methods. Considering the changes in physicochemical attributes, crystallization, and the IR spectrum, the formation of complexes can be assumed [25–28]. CD derivatives have been developed with more favorable attributes like increased solubility, stability, and less toxicity. In ophthalmic formulations the hydroxypropyl-β-and γ-cyclodextrin and sulfobuthylaether-β-cyclodextrin are the most commonly applied derivatives, which are also official in European Pharmacopoeia. Studies on rabbit corneal epithelial cell-line showed non-toxic attributes after application of these types of CDs [29–32]. Recent approaches are shown on Table 2., where CDs are applied in ophthalmic formulations.

**Table 2.** Recent approaches to use CDs in ophthalmic formulations.

| API | CD Derivative | Formulation | Reference |
| --- | --- | --- | --- |
| Flurbiprofen | HPBCD | eye drop | [33] |
| Nepafenac | HPBCD HPGCD | eye drop | [34] |
| Amlodipine | HPBCD SBEBCD | eye drop | [35] |
| Dexamethasone acetate | HPBCD HPGCD | eye drop | [36] |
| Cyclosporine | HPBCD | insert | [37] |

### 2.2. Mucoadhesion

Application of polymers for prolonged ocular drug delivery is a common strategy. When the viscosity is increased, necessarily not to a high level, the residence time of the eye drop on the surface of the eye is prolonged without any side-effect, such as visual disorder or irritation. Mucoadhesive polymers are especially useful, because of the possible adhesion due to the interaction of polymer chains and the mucin layer of the tear film. It is defined as bioadhesion if the polymer chains are attached to the biological surface. Several theories are associated with the mechanisms of mucoadhesion. The wetting theory describes the effect of drop spreadability and wettability on the eye surface. According to the electrosatic theory, electron transfer is the mechanism of mucoadhesion. Adsorption theory is about primary and secondary chemical bonds between the polymer and mucus. In the case of high molecular weight polymers, the diffusion of polymer chains and glycoproteins of mucus can interpenetrate into each other creating an intermolecular net and mucoadhesion. This mechanism is also known as mechanical theory (Figure 4) [38–42].

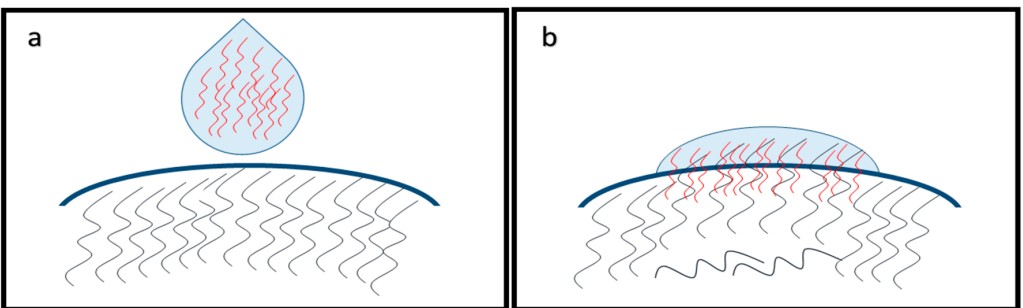

**Figure 4.** Mechanical theory of mucoadhesion. Before application of eye drop (**a**) and after interpenetration of polymer chains (**b**). After application, the polymer chains penetrate during the spreading of eye drop on the ocular surface.

The most commonly used mucoadhesive polymers are carbomers, alginates, methylcellulose, hydroxypropylmethylcellulose, hydroxyethylcellulose, chitosan, thiolated polymers and hyaluronic

acid. Biopolymers like hyaluronic acid are favorable in ophthalmic formulations, because of the biocompatible, non-toxic and biodegradable attributes [40,43,44].

### 2.3. Cyclodextrins and Mucoadhesive Polymers as Ophthalmic Drug Delivery Systems

Patient-compliance and effective therapy is desired, where the medicine is self-applicable, safe, and economical. Topical eye drop formulations would be optimal if they matched all previously mentioned requirements. Nowadays, several research groups have published articles about approaches for improved ocular drug delivery systems, where the API was dissolved in the aqueous solvent by CD complex formation, and the residence time was increased by mucoadhesive polymers, and therefore expected to have higher therapeutic efficacy [19,43,45].

Our research group has developed a promising formulation, where the prednisolone(PR)–CD complex was dissolved in aqueous solution containing zinc-hyaluronate and zinc-gluconate. With the addition of zinc-hyaluronate–zinc-gluconate system (ZnHA–ZnGlu), the antimicrobial stability was ensured during storage, application, and due to mucoadhesive attributes, increased residence time is expected on the eye surface. The osmolality and pH were set to physiological parameters by the sodium–chloride and borate buffer. Optimal PR diffusion was investigated in vitro using dialysis cellulose membrane. Mucoadhesive properties were tested by tensile test on mucin impregnated surface. All of the samples show that the mucoadhesivity, formulations with ZnHA–ZnGlu have significantly higher adhesive force values (Figure 5). [24].

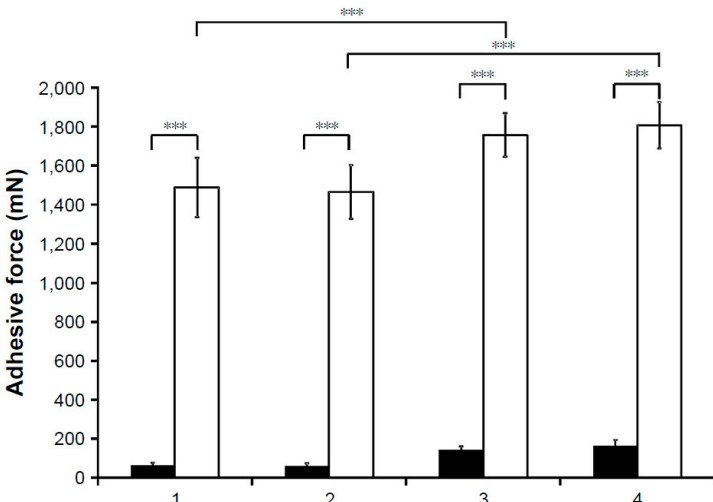

**Figure 5.** Results of tensile test with (white columns) and without (black columns) mucin. (1, with HPBCD; 2, with HPGCD; 3, with HPBCD and ZnHA–ZnGlu; 4, with HPGCD and ZnHA–ZnGlu) [24] (*** $p \leq 0.001$).

An innovative approach was published by Budai-Szűcs et al., where PR was incorporated into cyclodextrin-modified thiolated poly(aspartic acid) in aqueous in situ gelling solution. The complex formation was investigated by the XRPD method, as were the physicochemical attributes, rheology, and drug diffusion. In the drug diffusion study PR suspension was used as reference. The drug diffusion was tested over 24 h (Figure 6). In the case of the unbound PR cyclodextrin complex containing thiolated polymer gel, the diffusion rate was similar to PR suspension, due to the increased solubility and the prolonged effect of the polymer. In the formulation where the CD was covalently bound to the thiolated polymer, the drug diffusion was slower and dependent on the dissociation of PR from the CD complex and the inhibition of the polymer matrix. With the combination of the two type of formulations, an intermediate diffusion rate was observed. In this case the free PR–CD complexes caused a rapid biological effect, meanwhile the bound complexes prolong the drug release on the eye surface, therefore a less frequent application is needed to reach the target therapy [46].

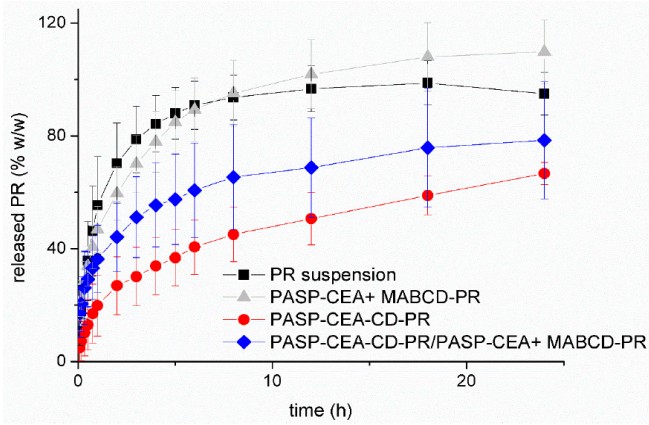

**Figure 6.** Drug release from the formulations containing prednisolone (PR). Cumulative mean values and standard deviations (SD) $n = 3$; PR: prednisolone, PASP-CEA: Thiolated poly(aspartic acid), MABCD: 6-monodeoxy-6-monoamino-beta-cyclodextrin hydrochloride [46].

Nanda et al. developed amlodipine containing mucoadhesive films through the addition of hydroxypropyl methylcellulose (HPMC) and CD derivatives (β-CD, HPBCD, sulfobuthylaether-β-cyclodextrin) using casting and solvent-evaporation methods. The authors investigated the swelling and erosive attributes, morphology, inclusion complex formation by DSC, FTIR and XRPD, in vitro and ex vivo diffusion of drug and anti-inflammatory effect on carrageenan induced rabbit model. As the result of the ex vivo permeations study shows, the applied CDs increased the permeability of drug on the excised sheep cornea, meanwhile the flux was dependent on the binding constants of the different CDs (Figure 7) [35].

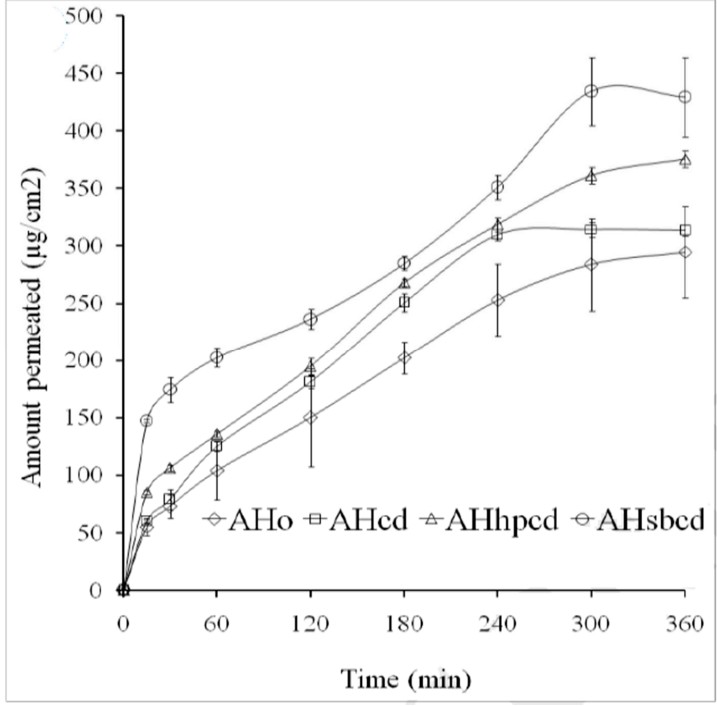

**Figure 7.** Ex vivo permeation study of amlodipine–CD containing HPMC films on excised sheep cornea. AHo: films without CD, AHcd: with β–CD, AHhpcd: With HPBCD, AHsbcd: With sulfobuthylaether-β-CD [35].

Shelley et al. published promising results about in situ gelling, nepafenac-HPBCD complex, sodium alginate containing ophthalmic formulation. Ex vivo permeability of API was tested using excised porcine cornea for 24 hours. The results showed that, the drug permeation from in situ gelling formulations was significantly higher compared with the official suspension formulation, Nevanac®, which was caused by the permeability enhancer effect of HPBCD. The highest permeation was observed in the case of composition F15 due to the low viscosity (Figure 8). This article also confirms the favorable effect of adding cyclodextrin and mucoadhesive polymers into the developed ocular drug delivery system [47].

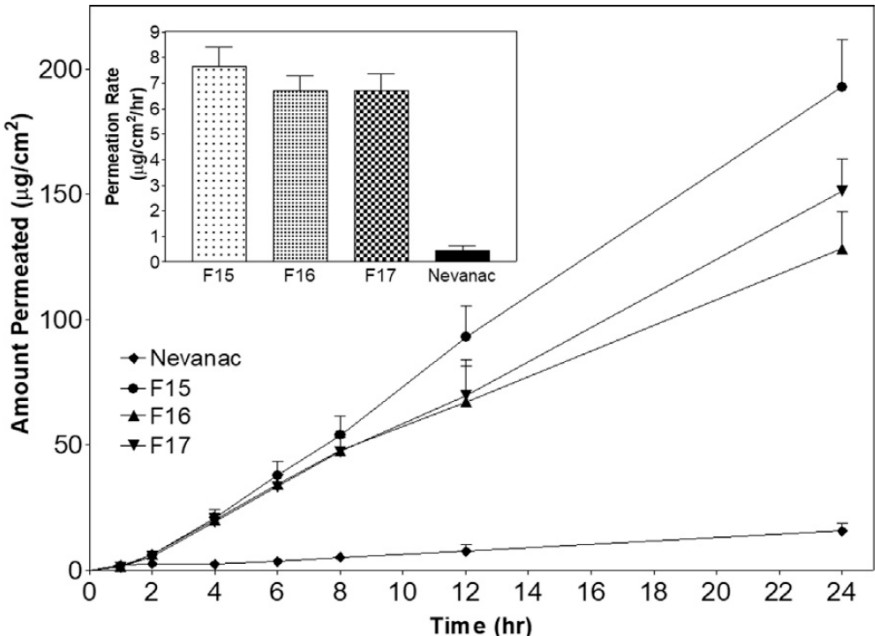

**Figure 8.** Ex vivo permeation test of nepafenac-HPBCD containing an in situ gelling system on porcine cornea. Nevanac: Official product used as reference, F15: 0.1 *w/v*% sodium-alginate, F16: 0.3 *w/v*% sodium-alginate, F17: 0.5 *w/v*% sodium-alginate [47].

## 3. Summary

In conclusion, the development of optimal ocular drug delivery systems is a major challenge in the pharmaceutical field. Ensuring proper therapeutic effect with acceptable patient-compliance are key considerations in the research of innovative formulations. Several promising approaches are described in the literature, where topical eye drops were characterized using cyclodextrins and mucoadhesive polymers together. Noticeably, there is lack of in vivo and clinical investigations, which are necessary to ensure innovative work is successful.

**Funding:** This research received no external funding.

**Conflicts of Interest:** The authors declare no conflict of interest.

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
