# Peer review of "Current Approaches to Use Cyclodextrins and Mucoadhesive Polymers in Ocular Drug Delivery—A Mini-Review"

_scipharm, doi:10.3390/scipharm87030015_

Reviewer 1 Report

Manuscript prepared by T.Bíró and Z. Aigner is devoted to review of approaches in which the cyclodextrins and mucoadhesive polymers are used in ocular drug delivery. Manuscript is well prepared. However, works by Prof. V. Khutoryanskiy are important in this field and should be mentioned. I think that these works could improve the quality of manuscript.

Author Response

Dear Reviewer,

We would like to thank you for the helpful suggestions. We have made the following changes in the manuscript:
The authors agree with the comment, that work of Prof. V. Khutoryanskiy is important in this field. The authors have completed the manuscript by the following references:
42. Khutoryanskiy, V.V. Advances in Mucoadhesion and Mucoadhesive Polymers. Macromolecular Bioscience 2011, 11, 748–764.
45. Morrison, P.W.J.; Connon, C.J.; Khutoryanskiy, V.V. Cyclodextrin-Mediated Enhancement of Riboflavin Solubility and Corneal Permeability. Molecular Pharmaceutics 2013, 10, 756–762.
The authors would like to ask for the acceptance of the corrections, and suggestion for publication.

06. 18. 2019.

Szeged, Hungary
Dr. habil Zoltán Aigner
corresponding author
associate professor
Institute of Pharmaceutical Technology and Regulatory Affairs
University of Szeged

Reviewer 2 Report

 I have some comments related to following aspects:

1. Some phrases are difficult to understand. For example, see lines 23-29, 36-38, 39-40, 47-49, 63-65, 73-75, … 174-177 and so on. Please reformulate them.

2. English revision is mandatory: see line 14 Addition of solubility enhancer cyclodextrin derivatives and mucoadhesive … I suggest By addition of solubility enhancer cyclodextrin derivatives and mucoadhesive … OR Adding of solubility enhancer cyclodextrin derivatives and mucoadhesive …

3. Please insert the abbreviation for Stability constant (line 121): Ks.

4. Please use patient-compliance instead patient-compliant.

Author Response

Dear Reviewer,
We would like to thank you for the helpful suggestions. We have made the following changes in the manuscript:
The authors confirm that some phrases are difficult to understand. The mentioned parts of the manuscript are reformulated considering the useful comments of the reviewer.
The authors would like to ask for the acceptance of the corrections, and suggestion for publication.

06. 18. 2019.

Szeged, Hungary

Dr. habil Zoltán Aigner
corresponding author
associate professor
Institute of Pharmaceutical Technology and Regulatory Affairs
University of Szeged